# Autism Spectrum Disorder and Duchenne Muscular Dystrophy: A Clinical Case on the Potential Role of the Dystrophin in Autism Neurobiology

**DOI:** 10.3390/jcm10194370

**Published:** 2021-09-24

**Authors:** Marta Simone, Lucia Margari, Francesco Pompamea, Andrea De Giacomo, Alessandra Gabellone, Lucia Marzulli, Roberto Palumbi

**Affiliations:** 1Biomedical Sciences and Human Oncology Department, University of Bari “Aldo Moro”, 70124 Bari, Italy; marta.simone@uniba.it (M.S.); fpompamea@gmail.com (F.P.); alessandragabellonee@gmail.com (A.G.); lucia.marzulli@uniba.it (L.M.); 2Basic Medical Sciences, Neurosciences, and Sensory Organs Department, University of Bari “Aldo Moro”, 70124 Bari, Italy; andrea.degiacomo@uniba.it (A.D.G.); roberto.palumbi@gmail.com (R.P.)

**Keywords:** case report, dystrophin, autism spectrum disorder, neurobiology, neurodevelopment

## Abstract

A diagnosis of autism spectrum disorder is reported in up to 19% of dystrophinopathies. However, over the last ten years, only a few papers have been published on this topic. Therefore, further studies are required to analyze this association in depth and ultimately to understand the role of the brain dystrophin isoform in the pathogenesis of ASD and other neurodevelopmental disorders. In this paper, we report a clinical case of a patient affected by ASD and Duchenne muscular dystrophy, who carries a large deletion of the dystrophin gene. Then we present a brief overview of the literature about similar cases and about the potential role of the dystrophin protein in the neurobiology of autism spectrum disorder.

## 1. Introduction

Dystrophinopathies are genetic disorders caused by mutations in the dystrophin gene located on the X chromosome [1]. The full-length dystrophin protein and the various shorter dystrophin isoforms are expressed in a tissue-specific manner in skeletal muscle, cardiac tissue, and the brain [2]. The three longest isoforms, Dp427M, Dp427C, and Dp427P, are expressed in skeletal and cardiac muscles, in the neurons of the cortex, and in cerebellar Purkinje cells, respectively. Mutations from exons 1 to 31 affect these isoforms [3]. Dp260 (mainly expressed in the retina), Dp140 (brain and kidney), and Dp116 (Schwann cells) are all involved by means of mutations between exons 31 and 62; Dp71, the most abundant brain isoform and also the shortest one, may be affected by the rarest mutation, that of downstream exon 63 [4]. Figure 1 schematically represents the dystrophin gene and isoforms [5].

Several authors reported frequent comorbidity between epilepsy and dystrophinopathies, suggesting that the absence of dystrophin might be related to increased central nervous system (CNS) excitability [6,7,8,9].

Indeed, some studies have demonstrated an increased prevalence of epilepsy in muscular dystrophy populations, ranging from 3.1% to 12.3% [10].

Hendrinksen et al. [6] demonstrated a high occurrence of epilepsy and other brain-related comorbidities in DMD. Furthermore, that study shows for the first time that the frequency of some of these disorders appear to be further increased when epilepsy is present next to DMD.

Moreover, many patients with dystrophinopathies have difficulties with communication and social behavior. Some studies have described a higher prevalence of autism spectrum disorder (ASD) in Duchenne muscular dystrophy (DMD) and Becker muscular dystrophy (BMD) compared to the general population, though the prevalence varied in these studies (3.1–20.7%) [11,12,13].

These clinical manifestations suggest the relevance of dystrophin, which might affect crucial neurodevelopmental pathways during childhood.

There is an increasing recognition that dystrophin plays a critical role in brain development and function, above all regulating the balance of synaptic excitatory and inhibitory neurotransmitter transmission. Indeed, Dp71 is involved in brain ion homeostasis, and its deficiency is expected to increase neuronal excitability, which might compromise the integrity of neuronal networks undertaking high-order cognitive functions [14].

However, the full role of dystrophin isoforms in the brain remains largely unclear; it is only recognized that the brain is affected by the lack of dystrophin and notably that mutations disrupting the brain isoforms Dp140 and Dp71 are more frequently associated with neurodevelopmental disorders [15].

Studies of genetic factors underlying the comorbid diagnosis of autism in associated genetic syndromes, such as DMD and milder BMD, have already opened routes into understanding the common biological mechanisms and intricate combination of multiple genes and mutations involved in the complex etiology of autism [16,17].

In a study published by Pagnamenta et al. 2011, starting from the Autism Genome Project Study, the authors described the data for a multiplex autism family presenting a rare copy number variant (CNV) of the DMD gene and a rare deletion involving the TRPM3 gene. The CNV led to a duplication in the DMD gene and this alteration was carried by two individuals of the family and their mother; the boys were diagnosed with ASD but they did not show any muscular manifestation [18].

A deeper analysis could lead to targeted interventions to improve neurodevelopmental outcomes. In our paper, we describe an illustrative clinical case concerning the overlapping conditions of ASD, febrile seizures (FS), and DMD. Then, we conduct a brief overview of the updated literature data about the potential role of the dystrophin protein in the neurobiology of autism spectrum disorder.

## 2. Case Description

The patient was the only child of two Italian healthy non-consanguineous parents.

A case of DMD was reported in the mother’s family. The patient was born at 38 weeks of gestation via spontaneous delivery; his neonatal period was characterized by physiological jaundice and weak suction.

At 5 months of age, during a hospitalization for an acute respiratory infection, a high value of creatine phosphokinase (CPK) was detected (13,023 U/L).

The genetic examination was conducted and revealed a large deletion including exons 1 to 44 of the dystrophin gene, predictive for DMD (PCR multiplex).

Developmental difficulties were reported in early childhood, involving the motor domain with the acquisition of independent walking at 30 months, and language and behavior.

At the age of 2 years and 8 months, the parents came to our attention because of the patient’s speech/motor delay, deficits in social communication and interaction, and restricted and repetitive behaviors. A comprehensive evaluation was conducted in the hospital. Neurological examination revealed hypotonia, a waddling gait, severe motor impairment, difficulty in postural passages with weakness of proximal muscles (Gowers sign), and hyporeactive osteotendinous reflexes. Communication skills included only vocalizations, with an impairment in language comprehension. The behavioral profile was characterized by deficits in social interaction, joint attention, eye contact, and motor stereotypes. The neuropsychological assessment was performed using the following standardized protocols: the Griffiths Scale for Mental Development [18,19], the Vineland Adaptive Behavior Scales—1st edition [20], the Psychoeducational Profile—Third Edition [21], and the Autism Diagnostic Observation Schedule—Second Edition (Module 1) [22]. The results obtained confirmed global developmental delay, social interaction, and behavioral impairment, suggestive of an ASD. Cardiologic, ocular, pneumologic, and ENT (otorhinolaryngologic) evaluations were conducted with unremarkable findings. CPK values were still high (14,130 U/L) and the electroencephalogram was normal. Therefore, we concluded with an ASD diagnosis (level 1 severity), associated with global developmental delay, in according with the Diagnostic and Statistical Manual of Mental Disorders—Fifth Edition (APA, 2013) [23].

The child was followed regularly and the data derived from the periodical assessment are summarized in Table 1.

At the age of 3 years and 8 months old, two episodes of febrile seizures occurred. These episodes were characterized by a generalized onset and brief duration and successive sleep electroencephalograms showed burst of frontal spikes tending to spread.

The last follow-up was conducted at the age of 5 years and 9 months old. The child, who was steroid-naïve, showed the persistence of impairment in the motor domain, the absence of language, and no acquisition of sphincter control. A standardized evaluation of nonverbal cognitive functioning was performed using the Leiter International Performance Scale—Revised (Leiter-R) [24], showing a non-verbal IQ of 65, suggestive of a mild intellectual disability. Biochemistry testing confirmed a high value of CPK (31,061 U/L; n.v. 31–152).

A diagnosis of ASD (level 2 severity), associated with mild intellectual disability, language impairment and repetitive and stereotyped behaviors was made according to the clinical evaluation and with the support of the standardized tests. A brain MRI was not performed because the child was not compliant, and the anesthesia might have been complicated by potential respiratory difficulties.

We proposed to the family that we could run a genetic panel for neurodevelopmental disorders with the aim of evaluating the presence of further potentially susceptible mutations.

## 3. Discussion

In this paper, we have described the case of a child affected by ASD, mild intellectual disability, DMD, and a history of febrile seizures, with a huge deletion of the proximal part of dystrophin gene.

While the co-occurrence of DMD and neurodevelopmental diseases is well known in literature, the reasons behind this association largely remain a mystery.

In the literature, there have been studies that have sought to correlate the location of the DMD gene mutation with the cognitive profile of affected individuals, revealing poorer cognitive performance in individuals carrying mutations distal to exon 45 in some cases [25]. This is in line with our finding of mild intellectual disability in the presence of a deletion in the proximal region of the dystrophin gene. However, other studies refute this hypothesis, suggesting that a clear genotype–phenotype correlation for cognitive impairment has yet to be established [26].

Similarly, how the position of the mutation of this gene could relate to autistic symptoms is not well known. In order to investigate the genotype–phenotype correlation in patients affected by dystrophinopathies and ASD, we conducted a literature search using a combination of the free-text terms “dystrophin” and “autism” on the Pubmed, Scopus, and Web of Science databases. We found that the studies carried out so far have obtained results that are not always concordant with each other. Some of them underlined the hypothesis that mutations in the distal part of the dystrophin gene, the expression of which is associated with the production of the shortest isoforms, such as Dp71, might be more associated with ASD than other mutations occurring in the upstream part of the gene [27,28]. The authors of a recent review, which examined in-depth the functions of the different dystrophin isoforms, found that Dp71 isoforms are differently expressed in the brain, they are located in specific sub-cellular sites, and they are also involved in different cellular functions such as membrane stabilization, cell division, intercellular adhesion, and importantly in synaptic organization [26].

On the contrary, other studies have reported that mutations involving a more proximal part of the gene were also described in ASD patients [12,13,29,30]. Ricotti et al. (2015) stated that upstream mutations of the gene, predominantly involving the Dp427 isoform, have been associated with cognitive and neurobehavioral problems, including the severity of autistic symptoms [12]. The authors hypothesized that these results might be associated with the expression of Dp427 in the GABAergic synapses in the hippocampus, cortex, and cerebellum. Additionally, they suggested that the absence of shorter isoforms is likely to have a cumulative effect on the loss of the full-length dystrophin products, adding neurobehavioral features to the neuromuscular symptoms. Concerning Dp140, it is highly co-expressed during brain development, along with other crucial proteins involved in neuronal migration, morphogenesis, and axon guidance [16,28]. Additionally, studies performed on DMD mouse models showed that Dp140 is also expressed by oligodendrocytes and that it might also be crucial for the myelinization process [29,30].

In addition, a recent study by Doorenweerd et al. (2017) reported on a complex analysis of the expression patterns of the dystrophin isoforms across developmental stages [16]. Interestingly, the authors found that the expression of Dp140 in the cerebral cortex is limited at the fetal age, whereas in the cerebellum it is also expressed during the following stages. Dp71, which is instead expressed in neurons and glia, is the most prevalent isoform in the brain in all developmental stages, except for the prenatal ones [16,26,30].

However, on the whole, the available literature suggests that ASD symptoms do not appear to be significantly related to mutations of a specific dystrophin isoforms.

It is known that, as in skeletal muscle, dystrophin is an important component of a protein complex that anchors intracellular cytoskeleton elements with the extracellular matrix in the brain tissue [27]. It is worth pointing out that other membrane proteins involved in the same functions as those of brain dystrophin have also been proposed for ASD neurobiology. There is some evidence that the scaffolding protein SHANK3 [31,32,33,34,35] or the neurexin/neuroligin proteins (pre- and post-synaptic proteins crucial for correct synaptic organization/function) might be involved in ASD neurobiology [36,37,38].

Furthermore, a possible role of dystrophin-associated proteins, DAPs, or dystrophin-related proteins, DRPs, in the association of ASD and DMD has been also proposed, but it has been poorly investigated to date. Wang et al. reported the case of a 7-year-old female patient with ASD associated with intellectual disability, epilepsy, and a short stature, carrying a duplication of a large region (18q12.1) involving the DTNA gene; this gene encodes for the α-dystrobrevin protein, suggesting its potential role in ASD pathogenesis [39].

A diagnosis of ASD and/or autistic features had been reported in two other patients with 18q chromosomal aberrations, which overlapped with the genomic region disrupted in Wang’s case for the encompassment of the DTNA gene [40,41]. The association between ASD and abnormalities involving the α-dystrobrevin gene may lie in the fact that this protein, which belongs to the dystrophin complex, seems to intervene in neuronal plasticity mechanisms. Concerning DRPs, two whole-exome sequencing studies suggested a potential association between ASD and mutation in the DRP-2 gene, which encodes for a dystrophin-related protein that seems to play a role in regulating the myelination of Schwann cells [42,43]. However, literature on this topic is scarce, with few and predominantly empirical studies; thus, at the current state of research, their results are to be considered preliminary.

In summary, studies suggest that dystrophin might play a critical role in brain development and potentially in the neurobiology of neurodevelopmental disorders. In particular, the neurobiology of ASD is extremely complex. The literature shows that several pathways are involved in the etiopathogenesis of this disorder, in which environmental triggers act on genetic vulnerability. As a result, some brain developmental pathways are altered, such as oxidative stress, inflammatory and immune processes, cell migration, the synaptogenesis, as well as glutamate–GABA equilibrium [44,45].

Interestingly, the clinical case that we describe in this paper is a report of a patient affected by DMD, ASD, and febrile seizures. Even if our patient has not yet undergone a complete genetic panel to evaluate the presence of further potentially susceptible mutations, this case might support the hypothesis of the connection between the loss of brain dystrophin and a condition of hyperexcitability state/seizure vulnerability that it might be linked to the homeostasis of the excitatory/inhibitory balance (E/I balance) in the central nervous system during its development.

Along with other etiopathogenetic hypotheses for ASD, the involvement of the E/I balance has been hypothesized already in the neurobiology of this disorder, also considering the high prevalence of the comorbidity between epilepsy and ASD [46,47]. The E/I imbalance theory in ASD, firstly proposed in 2003 by Rubinstein et al. [48], is based on the hypothesis that ASD might be linked to abnormal glutamatergic and GABAergic neurotransmission in key regions for ASD, such as the prefrontal cortex, amygdala, cerebellum, or striatum [48,49]. The E/I imbalance might be the consequence of an excessive activation of glutamatergic neurons (e.g., mediated by a dysfunction in glutamatergic receptors) or the result of abnormal GABAergic inhibitory activity (e.g., abnormal migration and/or maturation of the GABAergic neurons) [50,51]. A condition of brain hyperexcitability in DMD is also supported by a recent case report, describing a patient affected by DMD and West syndrome (WS), suggesting WS as a part of the neuropsychiatric syndrome associated with DMD (the third case of Duchenne muscular dystrophy and West syndrome) [52]. Hendriksen et al. reported that the loss of functional dystrophin might increase seizure vulnerability and/or epileptogenesis in DMD patients [26]. In conclusion, there is growing interest in and evidence on the potential role of dystrophin in brain development and consequently in ASD neurobiology. However, it is important to highlight that further genetic investigations should be conducted in patients with DMD and ASD and/or seizures in order to verify that their association is due to dystrophin gene mutation or due to different, separate DNA changes. It is, indeed, a very complex topic and further studies are needed to analyze to what extent brain dystrophin might be involved in neurodevelopmental pathways and how it might be potentially considered as a new biomarker for ASD.

## Figures and Tables

**Figure 1 jcm-10-04370-f001:**
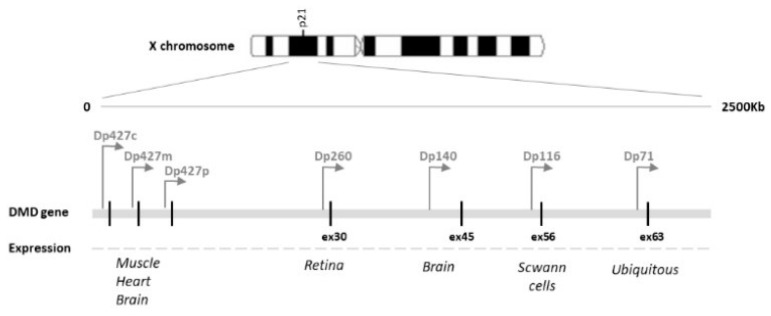
Schematic representation of the dystrophin gene and isoforms [5].

**Table 1 jcm-10-04370-t001:** Clinical evaluation of the patient.

CLINICAL EVALUATION					
	First Evaluation	Follow-Up 1	Follow-Up 2	Follow-Up 3	Follow-Up 4
*Age of the patient*	2 years 8 months	3 years 2 months	3 years 10 months	4 years 10 months	5 years 9 months
Developmental/cognitive evaluation					
**GSMD** **(age-related scores)**					
Motor scale	22.5 months	24 months	19.5 months	21 months	
Social skills	17 months	17 months	16.5 months	22.5 months	
Language	13 months	11 months	14 months	11 months	
Coordination	19 months	18 months	20 months	20.5 months	
Performance	13.5 months	15 months	21 months	21 months	
Total	17 months	17 months	18 months	18 months	
**Leiter-R**					
Total IQ					65
Adaptive functioning					
**VABS** **(age-related scores)**					
Communication	<1 year 6 months		<1 year 6 months	<1 year 6 months	
Daily living skills	1 year 8 months		1 year 8 months	1 year 8 months	
Social skills	1 year 7 months		1 year 8 months	1 year 6 months	
Motor skills	1 year 7 months		1 year 8 months	1 year 6 months	
Autism Spectrum Disorder					
**ADOS-2 (module 1)**					
Social Interaction score	14				
Stereotype behaviors and restricted interest score	2				
Total score	16				
Level of severity of symptoms	Mild				
**PEP-3**		NA	NA	NA	
*Performance subtests*	*Level of development*				
Verbal/preverbal cognitive	Severe (<12 months)				
Expressive language	Severe (<12 months)				
Receptive language	Moderate (<12 months)				
Fine motor	Severe (<12 months)				
Gross motor	Severe (<12 months)				
Visuo-motor imitation	Severe (12 months)				
Emotional expression	Moderate				
Social reciprocity	Moderate				
Characteristic Motor Behavior	Moderate				
Characteristic Verbal Behavior	Severe				
*Composite subtests*					
Communication	Severe (<12 months)				
Motricity	Severe (15 months)				
Impairment in adaptive behavior	Severe				

ADOS: Autistic Diagnostic Observation Schedule; GSMD: Griffiths Scale for Mental Development; ID: intellectual disability; IQ: intelligence quotient; NA: not available; PEP-3: Psycho-Educational Profile—Third Edition; VABS: Vineland Adaptive Behavior Scales.

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
