# Peer review of "Autism Spectrum Disorder and Duchenne Muscular Dystrophy: A Clinical Case on the Potential Role of the Dystrophin in Autism Neurobiology"

_jcm, 2021, doi:10.3390/jcm10194370_

Round 1

Reviewer 1 Report

typo line 153.  Ole should be "old"

Aprreciate that the authors have carefully addressed all the reviewer concerns.  

my only other concern is around the presentation of febrile seizures in this child.  Did the authors state that there was no other family history of febrile seizures (which can run as a familial disorder.   Was there any other genetic testing done in this child to confirm that there were no other reasons for the autism and febrile seizures?  

I do see that there was PCR testing for DMD deletion but not sure if other ASD related or Brain related testing was done.  In this modern era, one should verify that the association of DMD and ASD was not due to two separate DNA changes (and I suspect that there truly is an association between DMD and ASD) but that there is also no other genetic reason for the febrile seizures.

If this has been done, can the authors please note so? 

Author Response

Dear Reviewer,

here our answers to your helpful comments.

Typo line 153.  Ole should be "old"

Thank you. Done.

Appreciate that the authors have carefully addressed all the reviewer concerns. 

My only other concern is around the presentation of febrile seizures in this child.  Did the authors state that there was no other family history of febrile seizures (which can run as a familial disorder.   Was there any other genetic testing done in this child to confirm that there were no other reasons for the autism and febrile seizures? 

Thank you, we really appreciate this comment.

Not already done, but we informed the family about the possibility to perform a genetic investigation for neurodevelopmental disorders with the aim of evaluating the presence of further potentially susceptible mutations. We added this clarification also in the manuscript revised.

I do see that there was PCR testing for DMD deletion but not sure if other ASD related or Brain related testing was done.  In this modern era, one should verify that the association of DMD and ASD was not due to two separate DNA changes (and I suspect that there truly is an association between DMD and ASD) but that there is also no other genetic reason for the febrile seizures.

If this has been done, can the authors please note so?

Not done, please refer to the other answer.

Reviewer 2 Report

The manuscript Autism Spectrum Disorder and Duchenne Muscular Dystrophy: a clinical case on the potential role of the dystrophin in
autism neurobiology describes an interesting case of a DMD patient with ASD, general developmental delay and febrile seizures. The patient was followed from an early age with multiple follow-up assessments. The authors proceed to incorporate elements from the literature towards generating a hypothesis on excitatory and inhibitory (in)balance as a potential mechanistic pathway leading up to the reported symptoms.

The case itself is well presented, comprehensive and clear. However, I have some concerns regarding the discussion and the inclusion (and exclusion) of biological mechanisms from the ASD field. 

My main objection is that one of many hypotheses of the biological mechanisms involved in ASD is selected. That there are in fact more is not well described, nor is are there reasons provided why those were discarded. It would help to have a description of the literature search criteria in the review phase of the manuscript to better understand the selection process. The incorporation of the main hypotheses of biological pathways involved in ASD should be included. It is currently biased towards E/I balance. An equally compelling case could be made for the hypothesis of DNA methylation or the hypothesis of endocrine disruptors or neurogenesis, neuronal migration and morphogenesis (strong evidence for Dp140 to be involved in this), astrocytosis etc.. It is very complex and I am concerned the authors are oversimplifying this. The body of evidence from the case reports is insubstantial to select one over the other at this stage and I fully agree with the authors initial statement in line 12 that further studies are required to analyze this association in-depth and ultimately to understand the role of the brain dystrophin isoform in the pathogenesis of ASD and other neurodevelopmental disorders.

My other main comment is that the rationale of highlighting Dp71 was not clear to me. While it is indeed the highest expressed isoform after birth, it is only affected in 1-2% of patients, whereas the incidence of comorbidities is much higher. There is substantially more scientific data on Dp71 compared to the other isoforms in brain, but in relation to the case described it does not make sense to focus on this isoform. In the case described Dp71 is unaffected (theoretically without a brain biopsy). Could the authors explain why they propose a key role for Dp71, when especially in neurodevelopmental disorders early fetal development is a crucial phase, which happens to coincide with when Dp140 is dominant?

Minor comments:

Line 52 “above all regulating the balance of synaptic excitatory and inhibitory neurotransmission” requires referencing. This statement should indicate which isoform it relates to, which I expect to be full length. The different isoforms have separate function and are expressed in neurons, glia and pericytes respectively. As mentioned in the next sentence, the shorter isoforms are more frequently linked to learning and behavioural problems, so without specifying the isoform the statement may be misleading and suggest all dystrophin.

Line 61 recommend to include ref 26: demonstrated the strong significant genetic link in humans between the expression of the DMD gene and genes know to be associated with autism. The current reference refers to mice.

Line 81, can you confirm whether the first exon of isoform Dp140 in intron 44 was intact or included in the deletion?

Table 1, please indicate the unit of follow-up 4 Leiter-R (other items are age related scores, what is this item and what is the maximum?)

Author Response

Dear reviewer,

here you can find our replies to your very helpful comments:

The manuscript Autism Spectrum Disorder and Duchenne Muscular Dystrophy: a clinical case on the potential role of the dystrophin in autism neurobiology describes an interesting case of a DMD patient with ASD, general developmental delay and febrile seizures. The patient was followed from an early age with multiple follow-up assessments. The authors proceed to incorporate elements from the literature towards generating a hypothesis on excitatory and inhibitory (in)balance as a potential mechanistic pathway leading up to the reported symptoms.

The case itself is well presented, comprehensive and clear. However, I have some concerns regarding the discussion and the inclusion (and exclusion) of biological mechanisms from the ASD field.

My main objection is that one of many hypotheses of the biological mechanisms involved in ASD is selected. That there are in fact more is not well described, nor is are there reasons provided why those were discarded. It would help to have a description of the literature search criteria in the review phase of the manuscript to better understand the selection process. The incorporation of the main hypotheses of biological pathways involved in ASD should be included. It is currently biased towards E/I balance. An equally compelling case could be made for the hypothesis of DNA methylation or the hypothesis of endocrine disruptors or neurogenesis, neuronal migration and morphogenesis (strong evidence for Dp140 to be involved in this), astrocytosis etc.. It is very complex and I am concerned the authors are oversimplifying this. The body of evidence from the case reports is insubstantial to select one over the other at this stage and I fully agree with the authors initial statement in line 12 that further studies are required to analyze this association in-depth and ultimately to understand the role of the brain dystrophin isoform in the pathogenesis of ASD and other neurodevelopmental disorders.

My other main comment is that the rationale of highlighting Dp71 was not clear to me. While it is indeed the highest expressed isoform after birth, it is only affected in 1-2% of patients, whereas the incidence of comorbidities is much higher. There is substantially more scientific data on Dp71 compared to the other isoforms in brain, but in relation to the case described it does not make sense to focus on this isoform. In the case described Dp71 is unaffected (theoretically without a brain biopsy). Could the authors explain why they propose a key role for Dp71, when especially in neurodevelopmental disorders early fetal development is a crucial phase, which happens to coincide with when Dp140 is dominant?

Thanks for your helpful comments.

As you requested, we edited our manuscript and modified substantially our "Discussion" section, according to your concerns and helpful suggestions that definitely improved our paper.  

About your first major concern, in this new version, we added a description of the literature search criteria and we included other biological pathways besides the E/I imbalance theory.

Moreover, we totally agree with you when you say that Dp71 is unaffected in our patient and, as you suggested, we revised our manuscript in order to describe better the involvement also of other isoforms. Indeed, as you said, it is a very complex topic and further studies are necessary to investigate more in-depth the potential role of the dystrophin in ASD neurobiology.

Minor comments:

Line 52 “above all regulating the balance of synaptic excitatory and inhibitory neurotransmission” requires referencing. This statement should indicate which isoform it relates to, which I expect to be full length. The different isoforms have separate function and are expressed in neurons, glia and pericytes respectively. As mentioned in the next sentence, the shorter isoforms are more frequently linked to learning and behavioural problems, so without specifying the isoform the statement may be misleading and suggest all dystrophin.

Thank you for this comment. We have now added in the text this clarification.

Line 61 recommend to include ref 26: demonstrated the strong significant genetic link in humans between the expression of the DMD gene and genes know to be associated with autism. The current reference refers to mice.

Thank you, we really appreciate. We have now added that reference in the text.

Line 81, can you confirm whether the first exon of isoform Dp140 in intron 44 was intact or included in the deletion?

Yes, we can confirm that it was included.

Table 1, please indicate the unit of follow-up 4 Leiter-R (other items are age related scores, what is this item and what is the maximum?)

Thank you. To be more clear we have now left only the total IQ score.

Round 2

Reviewer 1 Report

thank you for making the suggested changes.

This manuscript is a resubmission of an earlier submission. The following is a list of the peer review reports and author responses from that submission.

Round 1

Reviewer 1 Report

Comments from reviewer
The article ”A review on the potential dystrophin role in Autism Spectrum Disorder neurobiology” by Simone et al present important findings and contribution to the knowledge of ASD in Duchenne and Becker muscular dystrophy.

Line 27-28:
Epilepsy in dystrophinopathies are not reported by several authors but only a few. I agree with  the statement of Goodwin et al (ref no 7) where they suggests that epilepsy may be a rare associated feature in children with muscular dystrophy secondary to dystrophin deficiency. This is also supported by the fact that epilepsy is not even mentioned in the care guidelines by Bushby et al: The Diagnosis and Management of Duchenne Muscular Dystrophy, Bushby K et al. Part 1: Lancet Neurol. 2010 Jan; 9(1):77-93. Part 2: Lancet Neurol. 2010 Feb; 9(2):177-189. As these guidelines are developed by many experienced neurologists, the presence of seizures in the patient population should most certanly been known and described.
Materials and Methods
Line 52: The article would benefit from a figure clarifying the selected procedure of inclusion of the studies presented.
Case description
Line 89: The clinically presented case is interesting but the correlation with febrile seizure has probably no significance on developing the ASD. 
Febrile seizures are not the same as having epilepsy. The prevalence of febrile seizures in the general population is relatively common, affecting 2-5% of children (see for instans following reference: Febrile seizures, Janet L Patterson, Stephanie A Carapetian, Joseph R Hageman, Kent R Kelley Pediatr Ann. 2013 Dec;42(12):249-54). There may be a coincindence that a patient with DMD also have febrile seizure, ("double trouble"). In my clinical practice I´ve met several patients with DMD and ASD without any history of seizures and also some patients with DMD that have had febrile seizures before being diagnosed with DMD and none of these patients have later on developed any symtoms autistic symptoms. Once more, referral to the lack of febrile seizure in the care guidelines is crucial. So my recommendation is to describe that the patient had febrile seizures but to minimize the impact on developing ASD.
I also lack information regarding medication with cortocosteroids. Has the patient been treated or not, and if so, at what age is the medication initiatet and is there a correlation between the initiated medication and the presenting autistic symptoms? It would be of interest to clarify this as the many professionals working with DMD claims that the behavioural problems that patients with DMD present are caused by the treatment with corticosteroids which is however, not my opinion. My experience and professional opinion is that the behavioural problems could be excaggerated when treatment with corticosteroids is initiated but it is not causing the austistic symptoms. This is in line with the presented hypthesis in this article that dystrophin have a role in brain development and also in developing ASD.
Discussion
Line 149-150: I totally agree with the authors suggestion that dystrophin might play a critical role in brain development and potentially in the neurobiology of neurodevelopment disorders.
Conclusion
Line 194-195 The authors statement that ”besides the neuromuscular features, it is important to consider neurobehavioral outcomes in DMD patient” is a clinically important point of view.
References
None of the articles included in Table 1 is included in the reference list wich must be added.

Author Response

The article ”A review on the potential dystrophin role in Autism Spectrum Disorder neurobiology” by Simone et al present important findings and contribution to the knowledge of ASD in Duchenne and Becker muscular dystrophy.

Line 27-28:

Epilepsy in dystrophinopathies are not reported by several authors but only a few. I agree with  the statement of Goodwin et al (ref no 7) where they suggests that epilepsy may be a rare associated feature in children with muscular dystrophy secondary to dystrophin deficiency. This is also supported by the fact that epilepsy is not even mentioned in the care guidelines by Bushby et al: The Diagnosis and Management of Duchenne Muscular Dystrophy, Bushby K et al. Part 1: Lancet Neurol. 2010 Jan; 9(1):77-93. Part 2: Lancet Neurol. 2010 Feb; 9(2):177-189. As these guidelines are developed by many experienced neurologists, the presence of seizures in the patient population should most certanly been known and described.

Materials and Methods

Line 52: The article would benefit from a figure clarifying the selected procedure of inclusion of the studies presented.

Thank you for this important suggestion. We have now added a clarifying flow chart about that.

Case description

Line 89: The clinically presented case is interesting but the correlation with febrile seizure has probably no significance on developing the ASD. 

Febrile seizures are not the same as having epilepsy. The prevalence of febrile seizures in the general population is relatively common, affecting 2-5% of children (see for instans following reference: Febrile seizures, Janet L Patterson, Stephanie A Carapetian, Joseph R Hageman, Kent R Kelley Pediatr Ann. 2013 Dec;42(12):249-54). There may be a coincindence that a patient with DMD also have febrile seizure, ("double trouble"). In my clinical practice I´ve met several patients with DMD and ASD without any history of seizures and also some patients with DMD that have had febrile seizures before being diagnosed with DMD and none of these patients have later on developed any autistic symptoms. Once more, referral to the lack of febrile seizure in the care guidelines is crucial. So my recommendation is to describe that the patient had febrile seizures but to minimize the impact on developing ASD.

Thank you for the important suggestion. We have modified the case presentation to minimize the impact of febrile seizure in confirming ASD diagnosis (line 154-156), given that it was not in our intention to directly link the history of febrile seizure to ASD diagnosis but to underline the epileptic vulnerability in this kind of patient. 

I also lack information regarding medication with corticosteroids. Has the patient been treated or not, and if so, at what age is the medication initiated and is there a correlation between the initiated medication and the presenting autistic symptoms? It would be of interest to clarify this as the many professionals working with DMD claims that the behavioral problems that patients with DMD present are caused by the treatment with corticosteroids which is however, not my opinion. My experience and professional opinion is that the behavioral problems could be exaggerated when treatment with corticosteroids is initiated but it is not causing the autistic symptoms. This is in line with the presented hypothesis in this article that dystrophin have a role in brain development and in developing ASD.

Thank you for the interesting question. Given that the child did not have a severe motor impairment, he did not start a corticosteroids treatment, so we can certainly exclude their potential role in the child’s behavioral abnormalities.  We have added this information along the main-text of the case description (line 142-143).

Discussion

Line 149-150: I totally agree with the authors suggestion that dystrophin might play a critical role in brain development and potentially in the neurobiology of neurodevelopment disorders.

Thanks for your comment.

Conclusion

Line 194-195 The authors statement that ”besides the neuromuscular features, it is important to consider neurobehavioral outcomes in DMD patient” is a clinically important point of view.

Indeed it is. Thanks for your comment.

References

None of the articles included in Table 1 is included in the reference list which must be added.

Thank you. Now we have added those references.

Reviewer 2 Report

This is an interesting review on the increasingly studied role of dystrophin on autism spectrum disorder neurobiology. The authors were able to concisely summarize the data currently available on this topic. Some concepts, however, would need to be dealt with in more detail.

I have specific comments/suggestions/questions:

Case report

-  Table 2 - ADOS-2 (page 4-5): in the “stereotyped behaviors and restricted interest” section the score is 0. However, in the text (page 4, Lines 100-101) you describe restricted and repetitive behaviors. Obviously, in clinical practice, ADOS itself does not make a diagnosis and the clinical observation is central but I was wondering how present restricted and repetitive behaviors were for never having been observed during ADOS.-         

- Page 5, lines 130: how do you confirm the diagnosis of ASD and the level of severity? (if with ADOS please add in the Table2). How and why the severity of ASD features changed? 

Discussion section

  •          Page 5, lines 136-137, lines 13-140: please specify the references.
  •          Page 5, lines 141-143, please insert the reference; it could be interesting to describe more in-depth what the authors found in the mdx models (for example the evidence of context-specific changes in social behavior and communication, based on the different behavioral/emotional demands and the study of ultrasonic communication in mice as possible explanation of atypical communication skills in ASD) and which hypothesis about the correlation between ASD and DMD they proposed  studying ASD features in mdx mice.
  • Page 6, line 160: please specify the reference.
  • Page 6, lines 160-170: to better support the possible role of DP71 in the autism, it could be interesting to refer to other studies, in which the same altered cellular functions (synaptic organization, ion channels, membrane stabilization, etc), related to other known genes, have been identified to contribute to ASD neurobiology.
  • Page 6, lines 180-185: please briefly explain the hypothesis about the E/I balance in ASD (excessive excitatory neurotransmission, malpositioning of inhibitory synapses, paradoxical effect of GABA with an excitatory effect due to depolarizing action)

Author Response

This is an interesting review on the increasingly studied role of dystrophin on autism spectrum disorder neurobiology. The authors were able to concisely summarize the data currently available on this topic. Some concepts, however, would need to be dealt with in more detail.

I have specific comments/suggestions/questions:

Case report

Table 2 - ADOS-2 (page 4-5): in the “stereotyped behaviors and restricted interest” section the score is 0. However, in the text (page 4, Lines 100-101) you describe restricted and repetitive behaviors. Obviously, in clinical practice, ADOS itself does not make a diagnosis and the clinical observation is central but I was wondering how present restricted and repetitive behaviors were for never having been observed during ADOS.         

Thank you for this comment. We made a mistake reporting a wrong score in the table. Now we have corrected it.

Page 5, lines 130: how do you confirm the diagnosis of ASD and the level of severity? (if with ADOS please add in the Table2). How and why the severity of ASD features changed?

The level of severity was increased according to clinicians and in accordance with DSM-5 on the basis on the request of support in everyday life.

In fact, in the table 2 the adaptive behaviors of the child, measured with VABS, did not improve with his growing up (equivalent age remained approximatively the same across lifetime).

Discussion section

Page 5, lines 136-137, lines 13-140: please specify the references.

Thank you for this suggestion.

We have now added the references.

Page 5, lines 141-143, please insert the reference; it could be interesting to describe more in-depth what the authors found in the mdx models (for example the evidence of context-specific changes in social behavior and communication, based on the different behavioral/emotional demands and the study of ultrasonic communication in mice as possible explanation of atypical communication skills in ASD) and which hypothesis about the correlation between ASD and DMD they proposed  studying ASD features in mdx mice.

Thanks for this comment. As suggested, we described more in-depth what the authors found in their study and the hypothesis proposed about the correlation between ASD and DMD.

Page 6, line 160: please specify the reference.

Done

Page 6, lines 160-170: to better support the possible role of DP71 in the autism, it could be interesting to refer to other studies, in which the same altered cellular functions (synaptic organization, ion channels, membrane stabilization, etc), related to other known genes, have been identified to contribute to ASD neurobiology.

Thanks for this helpful suggestion. In the revised manuscript, we added more references to studies investigating the role of other genes involved in the same altered cellular function of the brain dystrophin and that have been already proposed to have a potential role in ASD neurobiology.

Page 6, lines 180-185: please briefly explain the hypothesis about the E/I balance in ASD (excessive excitatory neurotransmission, malpositioning of inhibitory synapses, paradoxical effect of GABA with an excitatory effect due to depolarizing action).

Thanks again for your suggestion. As requested, we added a brief explanation of the E/I imbalance theory in ASD.

Reviewer 3 Report

Comments:

This paper reviews evidence for a role of the dystrophin protein in the neurobiology by summarizing and citing literature which has investigated relationship between dystrophin and ASD in children with DMD.  Then a case is presented.

Intro:  this is concise.

I’m surprised that the authors are quoting literature that is 20 years old in discussing clinical seizures and DMD.  Are they really related?  Need more recent literature noting this association. 

In the review of the DMD ASD literature, Table 1-the authors cite two types of studies, original research and case studies.  Can the authors please cite how these are different? Do the authors mean retrospective chart reviews when they use the term case study?   The descriptions need to be more detailed, eg retrospective chart review study, or another specific study type.

It would be interesting to present a figure with the dystrophin gene and show visually what is described verbally in the chart.

I believe that the case presentation adds nothing to this article, since there is a lot of documentation of ASD and DMD already.  There is no comment on whether febrile seizures run in this family and it is misleading to associate them with the DMD.  Two febrile seizures doesn’t make a diagnosis of epilepsy

The discussion is interesting summarizing a couple of research investigations looking at various dystrophin mutations and their role in ASD using mouse models.

To conclude, this manuscript would be better written without the case review and with more indepth analysis of the existing clinical literature. 

I think the associations with seizures should be left out as that is a different story.

Author Response

Comments:

This paper reviews evidence for a role of the dystrophin protein in the neurobiology by summarizing and citing literature which has investigated relationship between dystrophin and ASD in children with DMD.  Then a case is presented.

 Intro:  this is concise.

I’m surprised that the authors are quoting literature that is 20 years old in discussing clinical seizures and DMD.  Are they really related?  Need more recent literature noting this association. 

Thank you for this important suggestion. Now we have expanded the introduction adding more recent references about the existing correlation between epilepsy and DMD.

In the review of the DMD ASD literature, Table 1-the authors cite two types of studies, original research, and case studies.  Can the authors please cite how these are different? Do the authors mean retrospective chart reviews when they use the term case study?   The descriptions need to be more detailed, eg retrospective chart review study, or another specific study type.

Thank you for your very appropriate question. Effectively, the term “case-studies” that authors had used to describe their study design could create some misunderstanding. All the included papers are descriptive original research each with its own hypothesis and statistical analysis. Some of them were retrospective chart reviews, some others were cross-sectional studies with continuative recruitment of patients referred to a clinic. We have modified the table 1 according to your suggestions, that allow us to improve our manuscript.

It would be interesting to present a figure with the dystrophin gene and show visually what is described verbally in the chart.

Thank you for this suggestion. Since we were not able to produce a good figure for technical issues, we kindly asked for the use of a clarifying picture from another paper, and we are still waiting for permission from the journal.

I believe that the case presentation adds nothing to this article, since there is a lot of documentation of ASD and DMD already.  There is no comment on whether febrile seizures run in this family and it is misleading to associate them with the DMD.  Two febrile seizures doesn’t make a diagnosis of epilepsy

Thank you for your interesting suggestions. It is not in our intention to directly associate DMD and febrile seizures, so in discussion (line 244-246) we underlined the role of febrile seizures only as illustrative of such epileptic susceptibly, and not of diagnosis of epileptic disease. 

The discussion is interesting summarizing a couple of research investigations looking at various dystrophin mutations and their role in ASD using mouse models.

Thanks. We really appreciate your comment.

To conclude, this manuscript would be better written without the case review and with more indepth analysis of the existing clinical literature. I think the associations with seizures should be left out as that is a different story.

We have understood your point of view, anyway as the reviewer 1 & 2 and the editor have appreciated the case description, requesting to give it emphasis, we decided to keep it in the paper.

Round 2

Reviewer 3 Report

thanks to the authors for making the suggested changes and their thorough reply to all concerns.

I think that the inclusion of a dystrophin proteins and/or gene could add to the understanding of the manuscript.  I agree that it is best to obtain a figure already created with permission.

No other changes

Author Response

Dear revisor,
according to your suggestion, we have added a figure of the dystrophin gene, with exons and isoforms, to our manuscript. The figure is extracted from an open-access article distributed under the terms and conditions of the Creative Commons Attribution license "Fortunato, F.; Rossi, R.; Falzarano, M.S.; Ferlini, A. Innovative Therapeutic Approaches for Duchenne Muscular Dystrophy. J. Clin. Med. 2021, 10, 820. https://doi.org/10.3390/jcm10040820", which we properly cited in the capture of the figure.